# Ecological filters shape arbuscular mycorrhizal fungal communities in the rhizosphere of secondary vegetation species in a temperate forest

Yasmin Vázquez-Santos[1,2☯], Silvia Castillo-Argüero[2☯]*, Francisco Javier Espinosa-García[3☯], Noé Manuel Montaño[4☯], Yuriana Martínez-Orea[2‡], Laura V. Hernández-Cuevas[5‡]

1 Posgrado en Ciencias Biológicas, Universidad NacionalAutónoma de México, Unidad de Posgrado, Circuito de Posgrados, Coyoacán, Mexico City, Mexico, 2 Facultad de Ciencias, Departamento de Ecología y Recursos Naturales, Universidad Nacional Autónoma de México, Investigación Científica, Coyoacán, Mexico City, Mexico, 3 Instituto de Investigaciones en Ecosistemas y Sustentabilidad, Universidad Nacional Autónoma de México, Antigua Carretera a Patzcuaro, Morelia, Michoacán, Mexico, 4 Departamento de Biología, División de Ciencias Biológicas y de la Salud, Universidad Autónoma Metropolitana unidad Iztapalapa, Iztapalapa, Mexico City, Mexico, 5 Instituto Tecnológico de Tlajomulco, Tecnológico Nacional de México, Tecnológico Nacional de México, Circuito Metropolitano Sur, Tlajomulco de Zúñiga, Jalisco, Mexico

☯ These authors contributed equally to this work.
‡ These authors also contributed equally to this work.
* silcas@ciencias.unam.mx

**Data Availability Statement:** All relevant data are within the manuscript and its Supporting Information files.

## Abstract

The community assembly of arbuscular mycorrhizal fungi (AMF) in the rhizosphere results from the recruitment and selection of different AMF species with different functional traits. The aim of this study was to analyze the relationship between biotic and abiotic factors and the AMF community assembly in the rhizosphere of four secondary vegetation (SV) plant species in a temperate forest. We selected four sites at two altitudes, and we marked five individuals per plant species at each site. Soil rhizosphere samples were collected from each SV plant species, during the rainy and dry seasons. Soil samples from the rhizosphere of each plant species were analyzed for AMF spores, organic matter (OM), pH, soil moisture, and available phosphorus, and nitrogen. Three ecological filters influenced the AMF community assembly: host plant identity, abiotic factors, and AMF species co-occurrence. This assembly consisted of 61 AMF species, with different β-diversity values among plant species across seasons and altitudes. Canonical correspondence analysis revealed that AMF community composition is linked to OM and available P and N, with only a few AMF species co-occurring, while most do not. Our study highlights how ecological filters shape AMF structure, which is essential for understanding how soil and environmental factors affect AMF in SV plant species across seasons and altitudes.

**Funding:** The author(s) received no specific funding for this work.

**Competing interests:** The authors have declared that no competing interests exist.

## Introduction

Arbuscular mycorrhizal fungi (AMF, Glomeromycota) are an essential part of the plant microbiome [1], and these are considered "keystone microbes" due to their critical role in the assembly of plant communities [2,3]. The AMF coevolved with plant roots approximately 450 million years ago, giving rise to arbuscular mycorrhiza (AM) [4]. AM is the most crucial symbiotic association widely distributed in terrestrial ecosystems as a key biotic interaction for ecosystem functioning and resilience due to its predominantly mutualistic function [4,5]. AMF occur in complex and diverse communities in temperate forest soils. Even though the structuring of microbial and plant communities has been studied, the interaction of biotic and abiotic filters that structure these communities has been poorly investigated [6,7].

AMF are obligate symbionts of plant roots strongly influenced by the unique conditions of the rhizosphere [1]. The rhizosphere is the soil zone surrounding plant roots, where physical, chemical, and biological properties are influenced by root-soil-microbe interactions [8]. This interaction zone acts as a biological filter in the structuring of AMF communities, because in it we find the phytochemical load of plants, as they release a wide variety of organic substances through the roots, such as carbon exudates, amino acids, organic acids, fatty acids, proteins, lipids, and secondary compounds [4,9,10]. These substances act as substrates and chemical signals in plant-microorganism recognition [1,11]. For example, AMF recognize their host plant by exuding strigolactones and flavonoids [12]. Thus, root exudates represent fundamental means to alter the composition of the rhizosphere-associated AMF community [13], as each host plant species has a specific exudate profile [14–16], which contributes to modulate the arbuscular mycorrhizal association [6].

Glomeromycota show conserved functional traits at the family level [17,18]. AMF species can differ in their carbon demand from the host plant, phosphorus translocation to the roots, carbon storage, and production of both root-associated and extraradical biomass [17,19]. For example, members of the Gigasporaceae and Acaulosporaceae families tend to produce more extraradical mycelial biomass in the soil, whereas members of the Glomeraceae family are extensive root colonizers [19]. Thus, AMF functional traits may influence their community assembly in the rhizosphere. Therefore, the interactions between AMF species and their host plants should be considered as another biotic filter that structures AMF communities.

The biotic assembly of the rhizosphere has led to the recruitment and selection of unique microbial taxa in the soil [20,21]. Among them are AMF, which show significant differences in terms of their spore's abundance, species composition, and activity when comparing different soils or host plant species [22–25]. However, anthropogenic disturbances exert a strong pressure and cause loss of microbial diversity due to the changes in soil conditions, leading to significant changes in the AMF community [26].

The rapid establishment of secondary vegetation (SV) plant species can maintain the structure and complex formation of interconnected mycorrhizal networks [27] and the local pool of AMF influencing the mycorrhizal community. In addition, the rhizosphere of these plant species allows the soil to store propagules and organisms, as well as chemical compounds and nutrient levels characteristic of past plant-soil-microbe interactions (soil memory), which influence the establishment of plant species corresponding to more advanced successional stages [26,28]. Furthermore, the AM association in SV plant species buffers the loss of stability and functionality of ecosystems [29,30], which explains the ability of some AMF species to act as functionally equivalent species [31].

Plant species can enrich soils differently and affect rhizosphere microbial communities, including AMF [32,33]. Although AMF species are not species-specific symbionts, different AMF taxa are known to be particularly associated with different host plant species or genera

[34,35]. This suggests a certain "selectivity" in the formation of the arbuscular mycorrhizal association, which may be closely related to the biochemical environment of the soil rhizosphere.

In addition to root exudate diversity, edaphic filters are recognized as one of the most important drivers of AMF diversity [32,36,37]. For example, in sites with strong seasonality of rain, such as temperate forests, the fluctuations in soil moisture and nutrient availability favor or limit the sporulation of AMF species [38]. Low richness and high abundance of AMF species spores have been reported during the dry season, because low soil water availability leads to hydric stress, and in response, some AMF species increase their sporulation [39,40]. During the rainy season, AMF richness increases due to variation in edaphic environment and plant biological characteristics that favor the sporulation of a high number of AMF species but with a low spore abundance [41,42]. Thus, these abiotic filters influence the mutual dynamics of AMF and host plants.

Understanding the differences in plant and fungal biological traits associated with the temporal variation of edaphic filters, is then critical for unravel the assembly of the rhizosphere-associated AMF community [6,43–45]. The aim of this study was to analyze the diversity and structure of the AMF community associated with the rhizosphere of four plant species of SV in a temperate forest, where ectomycorrhiza has traditionally been considered as the most important symbiosis. To understand the extent to which the host plant, AMF species interactions, and the abiotic environment influence these rhizosphere-associated AMF communities and to gain insight into the structuring of AMF communities, we collected soil samples from four different SV plant species at different altitudes: *Acaena elongata* (Rosaceae), *Ageratina glabrata* (Asteraceae), *Solanum pubigerum* (Solanaceae), and *Symphoricarpos microphyllus* (Caprifoliaceae). These plant species are phylogenetically distant and have different life histories and phenologies. We hypothesize that the structure of the AMF community will vary among the studied plant species depending on the season and edaphic conditions. We will consider some traits of the host plant species and the seasonal changes in edaphic conditions.

## Materials and methods

### Study area

This study was conducted in the temperate forest of *Abies religiosa* (H.B.K.) Schl. and Cham. (fir), in the Magdalena river basin (MRB), Mexico City, Mexico. The climate is temperate sub-humid with summer rains, with a mean annual temperature of 18˚C, mean annual precipitation of 1250 mm, and a thermal oscillation of less than 5˚C [46]. This forest exhibits a strong seasonality, with the rainy season occurring from May to October and the dry season from November to April. The predominant soil group is Humic Andosol, according to the World Reference Base for Soil Resources [47]. It is characterized as an acid soil dominated by allophanes, it is rich in organic matter, with a sandy-loam texture and a high capacity to fix phosphates on the amorphous mineral clay surface, which limits P availability for plants [48,49].

### Sites selection

Previous data on the differences in environmental factors along an altitudinal gradient in this area led us to establish four sampling sites distributed at two altitudinal intervals in the *Abies religiosa* forest for this study (Fig 1). Sites 1 and 2 were located at the highest altitude (3192 m a.s.l.–3254 m a.s.l.), while sites 3 and 4 were established at the lowest altitude (3027 m a.s.l.–3100 m a.s.l.). The criterion for selecting the sites was the co-occurrence of the four SV plant species here studied. In each site, we marked five individuals per plant species with reproductive structures, obtaining a total of 80 sampled individuals.

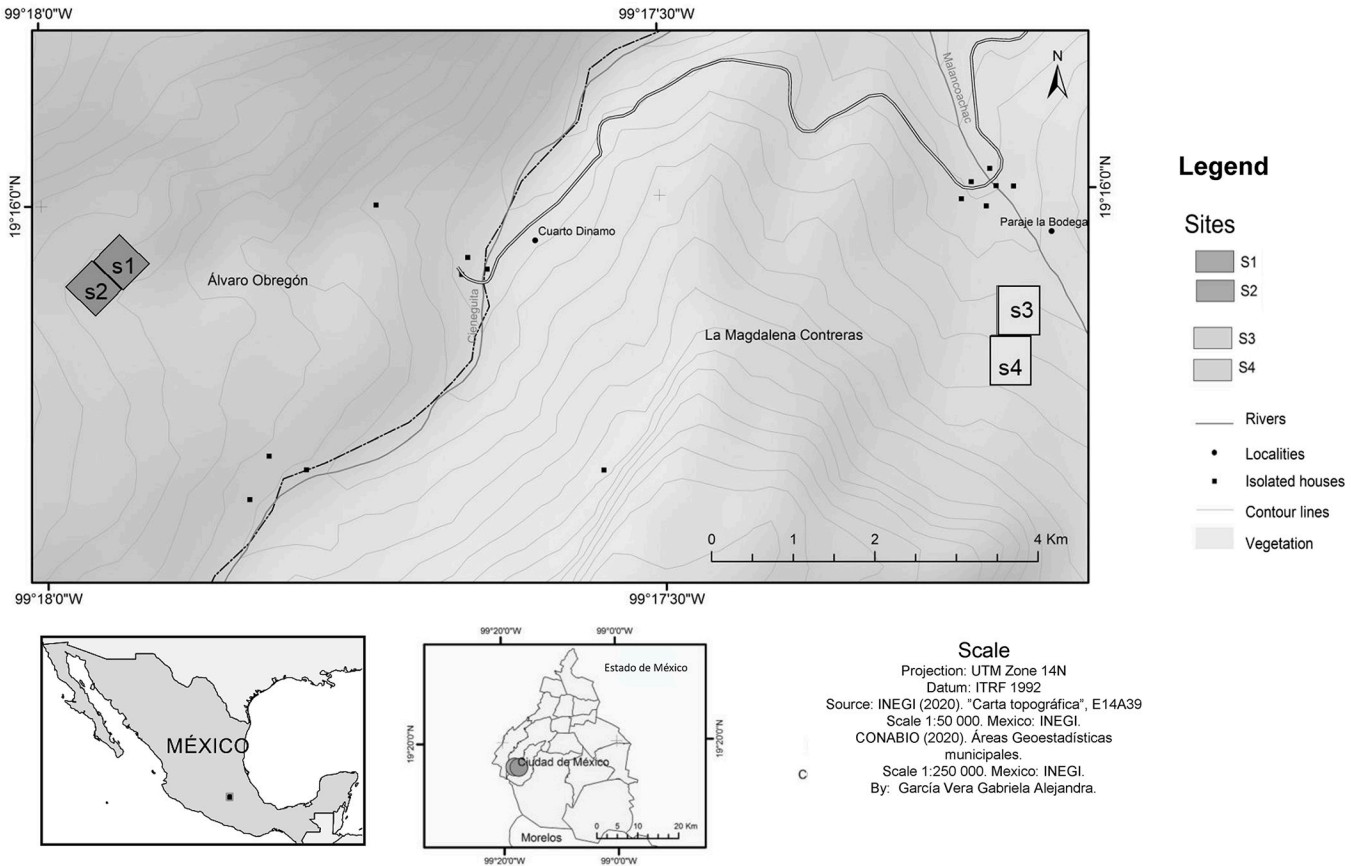

**Fig 1. Geographic location of the sampling sites.** *Abies religiosa* forest of the Magdalena river basin, Mexico City, Mexico. Site 1 = 2150 m$^2$, Site 2 = 2070 m$^2$, Site 3 = 2731 m$^2$, Site 4 = 2709m$^2$.

## Plant species selected

The selected plant species are abundant shrubs in the SV in the study forest [50]: *Acaena elongata* L., *Ageratina glabrata* (Kunth) R.M. King and H. Rob., *Solanum pubigerum* Dunal and *Symphoricarpos microphyllus* (Humb. and Bonpl. Ex Schult.) Kunth. They are abundant in the understory and found in natural regeneration sources, such as the seed rain and the seed bank [51]. The four plant species differ in flower and fruit types, seed size, and reproductive phenology patterns (Fig 2), [52].

## Sampling

We collected 150 g of soil from the rhizosphere of each marked individual. To ensure that the rhizosphere soil sample belonged to the selected individual, the surface soil and organic debris around the stem of each individual was removed to expose the root system. Soil samples were collected during the rainy season (August 2019) and the dry season (February 2020). Therefore, the total number of samples obtained was 160 (5 individuals × 4 plant species × 4 sites × 2 seasons). The soil samples were placed in labeled plastic bags and stored in the "Laboratorio de Dinámica de Comunidades" of the Faculty of Sciences, Universidad Nacional Autónoma de México (UNAM), where they were dried at room temperature. The local village authorities (comisariado) of "Magdalena Atlitic" gave us access to work and collect in their forests as part

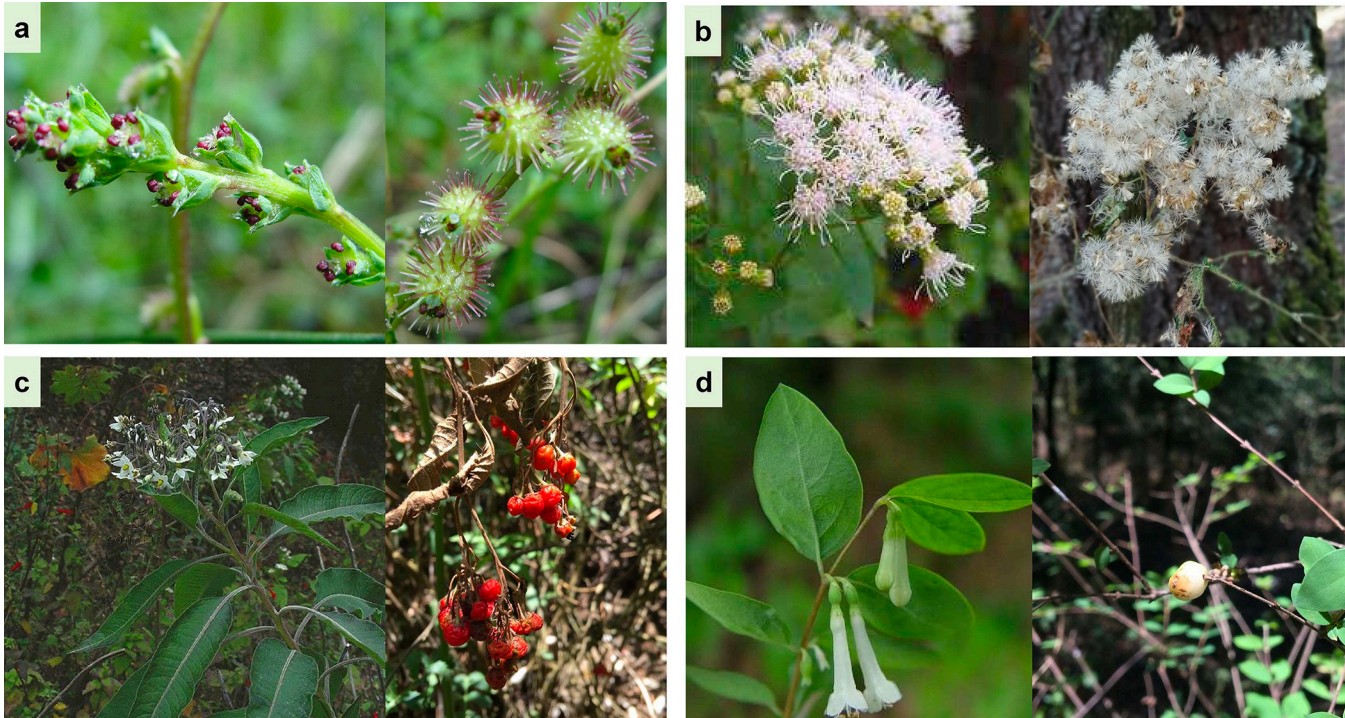

**Fig 2. Study plant species with their reproductive structures.** A) *Acaena elongata*, B) *Ageratina glabrata*, C) *Solanum pubigerum* and D) *Symphoricarpos microphyllus*, all located in the *Abies religiosa* forest of the Magdalena river basin, Mexico City, Mexico.

of the project "Efecto de los disturbios antrópicos en la diversidad functional en un bosque templado dentro de la Ciudad de México".

## AM fungi spores' extraction and species identification

We separated spores according to the method of [53], in 100 g of dry soil. Permanent slides were prepared according to the techniques of [53,54]. The AMF spores were separated into discrete groups using a dissecting needle according to their morphological characteristics of color and size. Half of the spores were placed in polyvinyl alcohol lactoglycerol (PVLG) and the other half in PVLG + Melzer's reagent.

The abundance quantification and taxonomic identification of the AMF spores were carried out in a microscope with Nomarski interference contrast (Nikon Optiphot II-Plus). There were only considered these with cytoplasmic content under the premise of viability. The morphological characteristics of the AMF spores (morpho-species) and the reaction of the wall layers with Melzer's reagent were considered for taxonomical identification. AMF species identification was carried out by comparing, and contrasting spores' features *versus* those published in specialized descriptions on genera and species of Glomeromycota, and taxonomical information available in the [55], http://invam.wvu.edu/the-fungi/classification, and the web site: [56] (http://www.zor.zut.edu.pl/Glomeromycota/), considering the taxonomical arrange proposed by [57,58]. The spores whose morphological characteristics agree with those described for a genus but did not agree with any previously described species in it, were designed as sp., and consecutively numbered. In addition, the spores designed as aff. were those whose morphology shown some variation in color's tones or sizes, regarding to description of the most similar species. The permanent slides of AMF spores were also preserved at

the collection of AMF spores in the "Laboratorio de Dinámica de Comunidades" of the Faculty of Sciences, UNAM, available for their observation.

## Abiotic factors

In each site, the monthly temperature was recorded by installing Hobos DATA loggers (easy-Log USB-ONSET, Massachusetts, EUA). The amount of light was measured by hemispherical photography using a Nikon camera (D80, USA) with a fisheye lens (EX SIGMA 4.5 mm f/2.8 DC HSM, USA) oriented towards the geographic north and positioned at ground level in the center of the site. The hemispherical photographs were analyzed using the Gap Light Analyzer program, ver. 2.00.

Additionally, in each site, 500 g of soil were also recollected from the first 0–20 cm deep in the mineral soil layer, this soil was analyzed for registering the following variables: (i) organic matter (OM) through a moist digestion [59]; (ii) relative soil moisture (RSM) was determined by the gravimetric method, where the soil was previously dried (105°C) until constant weight was obtained [60]; (iii) available phosphorus ($PO_4^-$) through a $NaHCO_3$ extraction 0.5 M to pH of 8.5 and colorimetric determination [61]; (iv) nitrates ($NO_3^-$) and ammonium ($NH_4^+$) were quantified through the use of a KCl solution and afterwards analyzed through chromatography [62]; (v) the pH value was determined with a potentiometer in a 1:2 ratio in deionized water (active acidity); (vi) Total nitrogen was determined by the Kjeldahl method which includes a sample digestion with sulfuric acid ($H_2SO_4$), steam distillation, neutralization; and (vii) Potassium (K) determination by atomic emission spectroscopy using NOM-02-SEMAR-NAT-2000 [63] techniques.

## Data analysis

Rarefaction curves were constructed for each plant species using the alpha diversity indices obtained by Hill numbers of order q0, q1, and q2. A total of one thousand resampling iterations (bootstrap or randomizations with replacement) were performed with 95% confidence intervals [64]. Hill numbers of order q0, which are insensitive to abundance, are synonymous with species richness [65]. Hill numbers of q1 order correspond to the exponential Shannon index, which reflects the diversity while taking into account the relationship between species abundance and number [65]. Hill numbers of order q2 represent the inverse of the Simpson index (1-D), which considers abundances and indicates species dominance and evenness per station. It quantifies the probability that of two randomly selected individuals within the sample belong to different species [66]. By using Hill numbers, we prioritize a balanced consideration of species richness and abundance, while reducing the disproportionate influence of very common or rare AMF species. This ensures a more equitable representation of species contributions to the ecological community, which is essential when assessing mycorrhizal diversity. Calculations were performed using the iNEXT (iNterpolation/EXTrapolation) package [64] in the R programming language, version 4.1.2 [67].

Additionally, we conducted a generalized linear mixed effects model (GLMER, Library lme4), [68] with a Poisson error structure to test for differences in the richness of AMF spore-based species, alpha diversity and spore abundance.

A species frequency matrix of AMF associated with the four host plant species was constructed. Because we don't have the security that there is a direct interaction between specific AMF and the plant species roots, we use the term of "associations" instead of "interactions" [69,70] throughout the manuscript. Based on this matrix, a bipartite interaction graph was generated, and network descriptors were determined [71]. At the network level, the degree of complementary specialization was obtained using the $H_2'$ index, which measures species

segregation. Values close to one indicate a high niche partitioning and high community-level specialization, while values close to zero indicate low levels of specialization [72,73]. The degree of network nestedness was quantified, represented by the NODF index. This index captures the relationship between specialists (species with few connections) and generalists (species with many connections) in the network. NODF values range from zero to 100, with high values indicating high nestedness and low values indicating low nestedness [74]. On the other hand, asymmetry determines the strength of the interaction between the two trophic levels and ranges from zero (no specialization) to one (maximum specialization). Additionally, the connectivity index (C) was obtained. Connectivity values close to 1 indicate perfect connectivity [75]. Finally, niche overlap was estimated. For this index, values close to zero indicate high niche partitioning, while values close to one indicate complete niche overlap [76]. Descriptors were calculated using the Bipartite package in R 4.1.2 [67].

Additionally, to understands the positive, negative, or random associations that occur between the AMF species associated with the rhizosphere of the marked plant individuals, a Probabilistic Species Co-Occurrence Analysis was performed using "cooccur" statistical package in R [77,78].

Beta diversity was assessed by dissimilarity in AMF species composition. A Permutational Multivariate Analysis (PERMANOVA) was performed to evaluate statistical differences in AMF species composition in host plant species, between seasons and altitudes, and their interaction. This analysis involved 999 random Monte Carlo permutations using the "Vegan" statistical package in R software, version 4.1.0. The PERMANOVA analysis tests for differences in variables within samples of different treatment groups [79]. In our study, it was used to assess variation in species composition among the four host plant species using the Bray-Curtis similarity index. The PERMANOVA analysis utilizes the T-statistic to determine the level of significance of separation between groups, with the associated P-value [79]. According to [79], the null hypothesis assumes that the centroids of each sample set (in the similarity space) are equivalent across landscape units. Therefore, if this hypothesis is rejected (based on the pseudo-F value), it indicates that the observed differences in centroids are greater than expected by chance alone [79]. In addition, a Non-metric Multidimensional Scaling (NMDS) was performed to visualize differences in AMF species composition between host plant species, also employing the Bray-Curtis dissimilarity index. NMDS was generated by using "metaMDS" in the "vegan" package. The matrix structure was constructed using AMF species abundance data, and 100 random iterations were performed using R 4.1.2 [67].

Finally, to analyzes the relationship among abiotic factors and composition and abundance of AMF associated with each host plant species, a Canonical Correspondence Analysis (CCA) was performed using PC-ORD software, version 7 [80]. We performed Variance Partitioning Analysis (VPA) to determine the specific and combined effects of biotic and abiotic factors on AMF community structure. Using the "vegan" package, along with CCA and permutation tests for significance, we delineated the influence of each factor.

## Results

### Alpha diversity

We recorded 61 AMF species belonging to 18 genera and nine families. The most abundant genera being *Acaulospora* (61%), *Ambispora* (6.1%), *Diversispora* (5.52%) and *Rhizophagus* (5.51%), and the least abundant ones were *Sclerocystis* (0.085%), *Scutellospora* (0.064%), *Entrophospora* (0.06%), *Gigaspora* (0.042%) and *Cetraspora* (0.02%). S1 Fig shows some AMF species associated to four SV plant species.

**Table 1. Richness, abundance, and diversity of AMF species registered in the rhizosphere of four secondary vegetation plant species of the *Abies religiosa* forest of the Magdalena river basin, Mexico City, Mexico.**

| Host plant | q0 | | q1 | | q2 | |
|---|---|---|---|---|---|---|
| | Observed | Estimator | Observed | Estimator | Observed | Estimator |
| *Acaena elongata* | 30 | 30 | 11.17 | 11.37 | 5.94 | 5.97 |
| *Ageratina glabrata* | 42 | 50.1 | 11.49 | 11.68 | 6.74 | 6.30 |
| *Solanum pubigerum* | 42 | 54.24 | 19.60 | 19.85 | 14.5 | 14.6 |
| *Symphoricarpos microphyllus* | 48 | 48.49 | 14.66 | 14.73 | 8.70 | 8.72 |

According to the rarefaction curves of each host plant species, we observed that the richness (q0) was stabilized only for *Symphoricarpos microphyllus* (S2 Fig). *Acaena elongata* obtained an effective number of AMF species of q0 = 30, *Ageratina glabrata* q0 = 42, *Solanum pubigerum* q0 = 42 and *Symphoricarpos microphyllus* q0 = 48 (Table 1). Each value reflects the total number of unique AMF species found for each plant species, with *Symphoricarpos microphyllus* having the greatest AMF species richness.

According to the GLM, season, plant species, altitude, and the interaction between these three sources of variation have a significant effect on AMF species richness (q0) and AMF spore abundance (Table 2). The number of different AMF species and the abundance of fungal spores present were influenced by the season, the diversity of plant species, and the altitude at which they occur. In contrast, these effects were not observed for q1 alpha diversity, which encompass the number of AMF species and their relative abundances.

## Interaction networks

Seasonal and altitudinal variations in the number of AMF species in the rhizosphere were observed. During the rainy season, the association network included 48 AMF species (out of 61 recorded) (Fig 3), while during the dry season this network contained 49 AMF species (Fig 3), with 36 species occurring in both seasons. The association network at the high altitude had 54 AMF species (Fig 3) and that at the low altitude had 51 AMF species (Fig 3), with 44 species shared.

We have recorded 21 generalist AMF species (35% of the total richness) and 15 exclusive AMF species (24.5% % of the total richness). The rest of the 25 AMF species have at least two plant species in common.

The common species represent a significant portion of the AMF community, indicating their widespread distribution at both altitudes. Network association parameters are presented

**Table 2. Summary of the statistical results of the Generalized Linear Mixed effects (GLM) for species richness and abundance of arbuscular mycorrhizal fungi (AMF).** Four plant species measured at two altitudes during the rainy and dry seasons.

| Source of variation | df | AMF Richness | | AMF Abundance Spores | |
|---|---|---|---|---|---|
| | | sd. Dev | $X^2$ | sd. Dev | $X^2$ |
| Species | 3, 116 | 258.93 | **<0.01** | 6186.5 | **<0.0001** |
| Season | 1,119 | 270.73 | **<0.01** | 6912.3 | **<0.0001** |
| Altitude | 1, 115 | 240.41 | **<0.0001** | 5821.9 | **<0.0001** |
| Species: Season | 3, 112 | 239.19 | 0.748 | 5484.2 | **<0.0001** |
| Species: Altitude | 3, 108 | 237.63 | 0.888 | 4819.2 | **<0.0001** |
| Season: Altitude | 1, 111 | 238.27 | 0.335 | 4908.3 | **<0.0001** |
| Species: Season: Altitude | 3, 105 | 220.49 | **<0.001** | 4766 | **<0.0001** |

* Significant differences are in bolds.

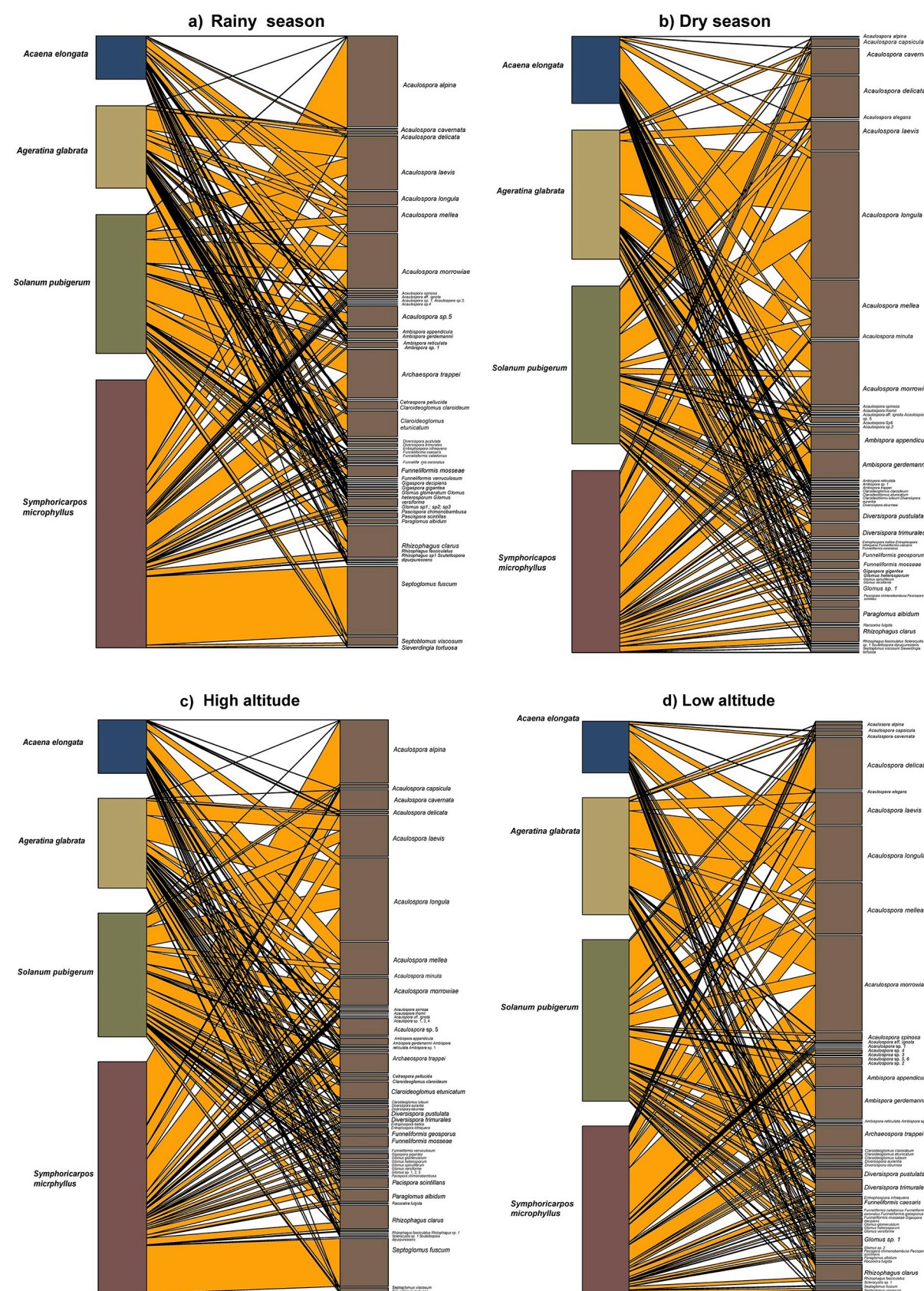

**Fig 3. Network of interactions between the host plant species and the arbuscular mycorrhizal fungi in both seasons and at different altitudes in an *A. religiosa* forest of the Magdalena river basin, Mexico City.** Rainy season and dry season (a and b), high altitude and low altitude (c and d).The lines represent connections, and their thickness indicates the intensity of these connections. The size of the nodes is related to the number and intensity of the connections.

in Table 3. The fact that not all species are present in each network, suggests that species are sensitive to and selectively respond to ecological variables such as seasonality and altitude, as well as to a partial niche overlap.

Sporulation of AMF species occurs mainly during the dry season. *Acaena elongata* hosted 25 AMF species during the rainy season, while 30 AMF species were recorded during the dry season. *Solanum pubigerum* recorded 25 AMF species in its rhizosphere during the rainy season, while 31 AMF species were found during the dry season. *Symphoricarpos microphyllus* recorded 31 AMF species in its rhizosphere during the rainy season, while 34 AMF species were found during the dry season (S1 Table). Among the four host-plant species studied, *Ageratina glabrata* showed a different pattern, with the highest number of AMF species in its rhizosphere during the rainy season (33-rainy and 30-dry).

We can visualize the host specificity of AMF through the presence of exclusive species. In the rhizosphere of *Acaena elongata*, we did not observe exclusive species, their 30 reported species are shared with at least one other plant species. For *Ageratina glabrata* six species were exclusive (14% of its total species richness): *Acaulospora thomii*, *Cetraspora pellucida*, *Claroideoglomus luteum*, *Entrophospora baltica*, *Rhizophagus sp2.*, *Sclerocystis sp. 1*. While *Acaulospora capsicula*, *Acaulospora elegans*, *Glomus spinuliferum* and *Gigaspora decipiens* were exclusive for the rhizosphere of *Solanum pubigerum* (9.5% of its total species richness). Finally, five AMF exclusive species were recorded for *Symphoricarpos mycrophillus* (10.4% of its total species richness): *Acaulospora minuta*, *Acaulospora* sp.1, *Acaulospora* sp. 3, *Funneliformis caledonius* and *Glomus* sp. 3 (S1 Table).

## Co-occurrence of AMF species

Among the 61 paired relationships of AMF species recorded in the rhizosphere soil of the 121 individuals sampled from the four host SV plant species, we observed that *Glomus* sp1 excluded 96% of the species, *Pascispora scintillans* a 95%, *Acaulospora capsicula* a 90%, *Claroideglomus claroideum*, *Glomus heterosporum*, and *Acaulospora spinosa* a 83.6%, the decreasing

**Table 3. Descriptive statistics of the plant–arbuscular mycorrhizal fungi (AMF) networks by season and altitude for four host plant species in the *Abies religiosa* forest of the Magdalena river basin, Mexico City, Mexico.**

|  | Season | | Altitude | |
|---|---|---|---|---|
|  | **Rainy** | **Dry** | **High** | **Low** |
| Connections per AMF species | 1.39 | 1.61 | 1.5 | 1.4 |
| Connections per plant species | 17.66 | 22 | 22.5 | 19.5 |
| $H_2$' | 0.40 | 0.29 | 0.31 | 0.29 |
| NODF | 60.59 | 64.18 | 64.66 | 65.98 |
| Connectivity of the network | 0.58 | 0.63 | 0.61 | 0.58 |
| niche overlap AMF | 0.33 | 0.43 | 0.45 | 0.47 |
| niche overlap plant | 0.43 | 0.46 | 0.42 | 0.41 |
| Modularity (Q) | 0.342 | 0.320 | 0.345 | 0.341 |
| Nestedness | 28.73 | 31.35 | 25.33 | 21.90 |

* $H_2$': Network specialization, NODF: Nestedness metric based on overlap and decreasing fill, Q: Modularity.

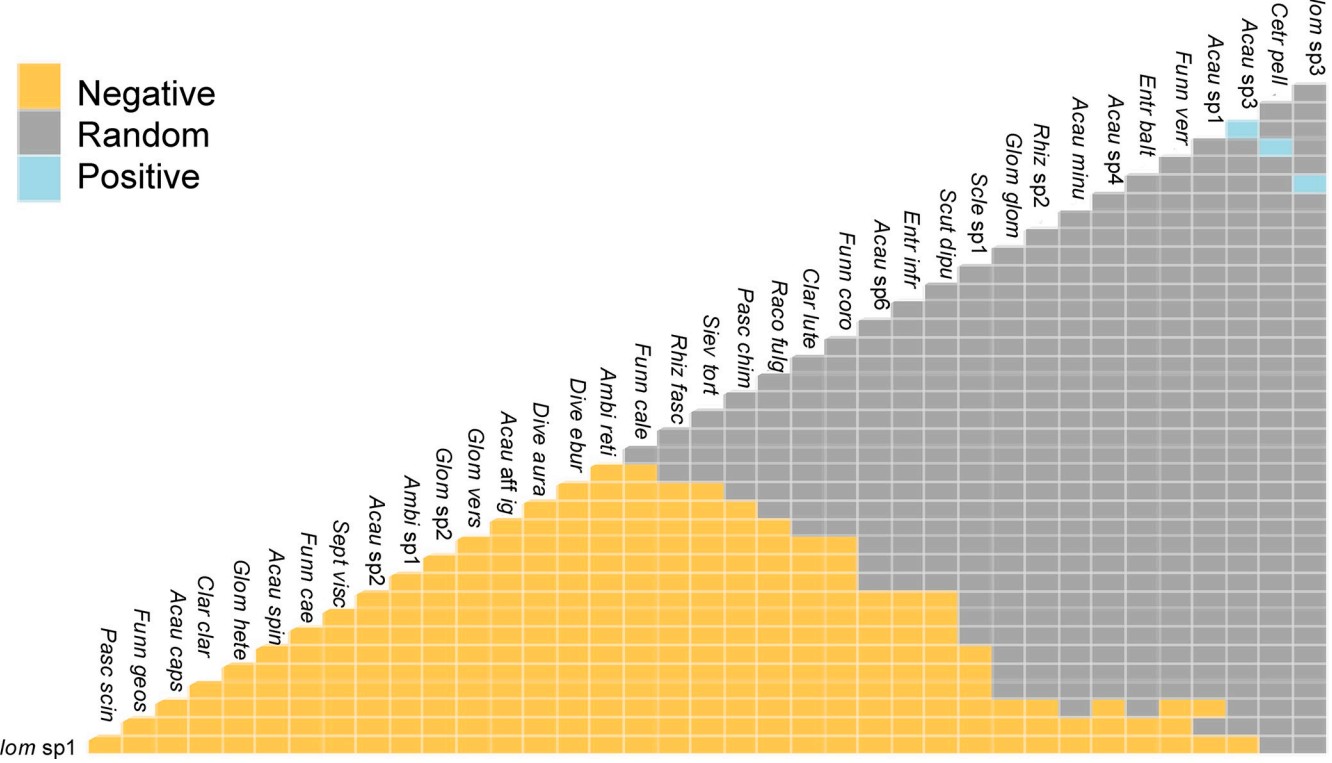

**Fig 4. Co-occurrence Matrix Species for all possible pairwise comparisons among arbuscular mycorrhizal fungi (AMF) species.** Map showing the positive and negative species associations determined by the probabilistic co-occurrence model for secondary vegetation four plant species in the *Abies religiosa* forest of the Magdalena river basin, Mexico City, Mexico. AMF species names are positioned to indicate the columns and rows that represent their pairwise relationships with other species.

gradient of exclusion ended with *Ambispora reticulata*, which excluded 27.86%. Of the AMF species (Fig 4). These AMF species are exclusive to at least one host plant species (S1 Table). Few species showed positive interactions.

## Beta diversity

Consistent with previous results the beta-diversity of the AMF community associated with the rhizosphere soil of the studied plant species is influenced by the seasonality, altitude, and the interactions among these variation factors, according to PERMANOVA (Table 4).

The non-metric multidimensional analysis (NMDS) was acceptable (stress = 0.202). A segregation of beta diversity was observed among the sampled individuals according to seasonality and altitude (Fig 5). Individuals of *Acaena* and *Ageratina* are more similar in the dry season, while *Solanum* shows similarity only at high altitudes in the dry season. *Symphoricarpos* individuals are consistently similar under all conditions. The NMDS of four plant species is shown in S3 Fig.

## Abiotic factors

We observed differences in the abiotic factors associated with temporality, altitude, and their interactions (Table 5). During the rainy season, soils at low altitude tend to be more acidic (5.9 ±0.001) with a higher content of organic matter (OM) (18.9 ±1.2%), which is even more pronounced at the high altitude. Soil moisture and nutrient levels are significantly higher during

**Table 4. PERMANOVA analysis with Bray-Curtis dissimilarity indexes with altitude and season factors for four plant species studied in the *Abies religiosa* forest of the Magdalena river basin, Mexico City, Mexico.** *d.f.* = degrees of freedom, R2 = variance explained by the factors, P = probability value.

| Source of variation | df | R$^2$ | F | P |
|---|---|---|---|---|
| Species | 3 | 0.07156 | 3.5464 | **0.001** |
| Season | 1 | 0.0566 | 8.414 | **0.001** |
| Altitude | 1 | 0.02554 | 3.7969 | **0.001** |
| Species: Season | 3 | 0.05743 | 2.846 | **0.001** |
| Species: Altitude | 3 | 0.03549 | 1.7589 | **0.001** |
| Season: Altitude | 1 | 0.01581 | 2.3507 | **0.005** |
| Species: Season: Altitude | 3 | 0.0313 | 1.5512 | **0.011** |

* Significant differences are in bolds.

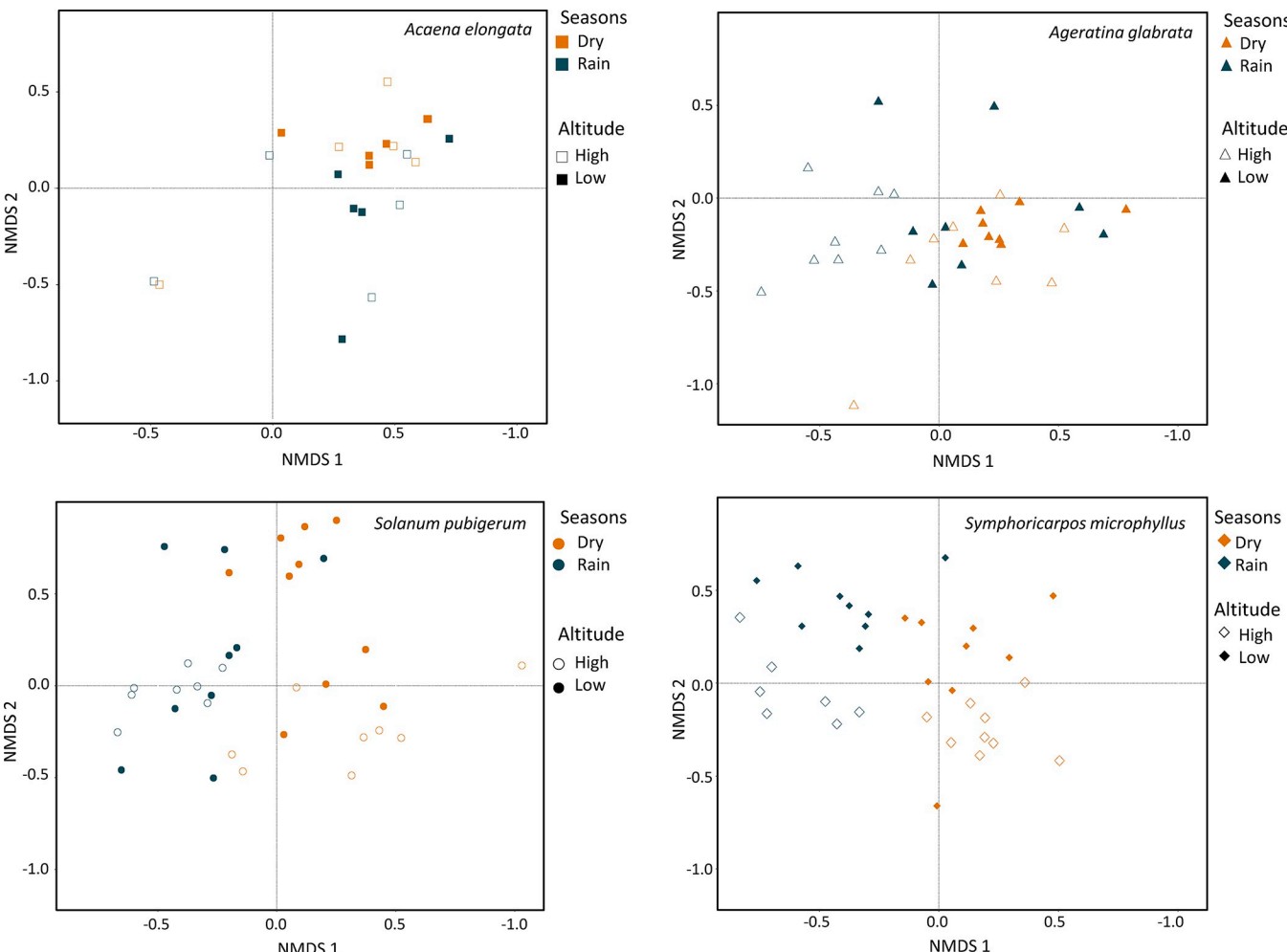

**Fig 5. Arbuscular mycorrhizal fungi (AMF) beta diversity.** Non-metric multidimensional scaling showing the dissimilarity in AMF species composition of sampled individuals (dots) of each host plant species in the wet and dry seasons and their location at high and low altitudes in the *Abies religiosa* forest of the Magdalena River basin, Mexico City, Mexico. A) *Acaena elongata*, B) *Ageratina glabrata*, C) *Solanum pubigerum* and D) *Symphoricarpos microphyllus*. The different symbol shapes are related to the host plant species and altitude, and the different colors indicate the season (bluish-green for the rainy and orange for the dry one).

**Table 5. *F*-ratios and p-values of a two-way ANOVA conducted on soil properties, light, canopy, and temperature in the *Abies religiosa* forest of the Magdalena river basin, Mexico City, Mexico.**

| Source of variation | | pH in $H_2O$ | | Electric Conductivity | | Organic matter | | Total Nitrogen | | Phosphorous available | | Potassium | |
|---|---|---|---|---|---|---|---|---|---|---|---|---|---|
| | *df* | F | P | F | P | F | P | F | P | F | P | F | P |
| Season | 1 | 335.7 | <0.001 | 345.2 | <0.001 | 78.1 | <0.001 | 12.7 | <0.001 | 178.3 | <0.001 | 8.2 | <0.01 |
| Altitude | 1 | 481.1 | <0.001 | 18.2 | <0.001 | 47.9 | <0.001 | 32. | <0.001 | 951.4 | <0.001 | 43.1 | <0.001 |
| Season: Altitude | 1 | 176.6 | <0.001 | 1.6 | 0.19 | 26.7 | <0.001 | 131.5 | <0.001 | 20.5 | <0.001 | 7.9 | <0.001 |
| **Source of variation** | | **Nitrates** | | **Ammonium** | | **RSM** | | **Light** | | **Canopy** | | **Temperature** | |
| | *df* | F | P | F | P | F | P | F | P | F | P | F | P |
| Season | 1 | 946.7 | <0.001 | 41.8 | <0.001 | 150.4 | <0.001 | 0.8 | <0.001 | 1.7 | <0.001 | 551.2 | <0.001 |
| Altitude | 1 | 11.7 | <0.001 | 115 | <0.001 | 31.1 | <0.001 | 28.9 | <0.001 | 55.6 | <0.001 | 30.2 | <0.001 |
| Season: Altitude | 1 | 13.8 | <0.001 | 54.4 | <0.001 | 0.8 | 0.36 | 4.4 | 0.035 | 2.9 | 0.085 | 1.04 | 0.308 |

\* Significant differences are in bolds.

this season: RSM (70.82 ±4.53%), N (0.6 ±0.04%), $PO_4^-$ (11.46 ±4.42 ppm), $K^+$ (0.88 ±0.10 ppm), and $NO_3^-$ (63.05 ±8.45 mg kg$^{-1}$). The RSM showed the highest values at the low altitude (66.74 ±11.06%). Conversely, during the dry season, there is increased light availability and canopy openness at the low altitude (16.5 ±5.02 Mols m$^2$s$^{-1}$, 10.21 ±3.05%, respectively). Finally, temperature was high during the rainy season (10.15 ±1.32°C) at the low altitude.

### Relationship between AMF diversity and abiotic factors

The CCA analysis revealed the interrelationship between biotic and abiotic filters in the incidence of AMF species. The composition and abundance of the AMF species associated with the rhizosphere of the four plant species vary depending on the physical conditions of the environment, seasons, and altitudes (Fig 6). According to the Monte Carlo permutation test (S2 Table), this pattern was found to be statistically significant.

During the rainy season the factors that determined the groups at the high altitude were total N, available P, OM, and nitrates in the soil with an AMF composition species in particular. In contrast, at the low altitude, higher RSM, warmer temperatures and soil electrical conductivity were strongly associated with a distinctive AMF community composition. The patterns revealed by CCA are consistent with the two-way ANOVA (Table 5) and with the variation in edaphic and climatic factors associated with elevation.

During the dry season and at the high altitude, soil pH may be one of the main factors influencing AMF community structure. In contrast, at the low altitude, we found that light intensity canopy openness, and soil ammonium were the determining factors.

Comparing the relative importance of the community structuring filters considered separately (Variance Partition Analysis, S4 Fig), we found that biotic factors accounted for 10% and abiotic factors for 8% of the variation in AMF community structure, with an additional 4% attributed to their interaction.

### Discussion

This research focused on assesing of AMF communities in the soil rhizosphere of four plant species that are abundant and characteristic of a temperate secondary forest in central Mexico. Our results show the ecological importance of SV plant species as potential drivers of local AMF diversity (spore bank) and suggest to three interacting primary filters that modulate species composition in AMF communities in the rhizosphere through time and space. These

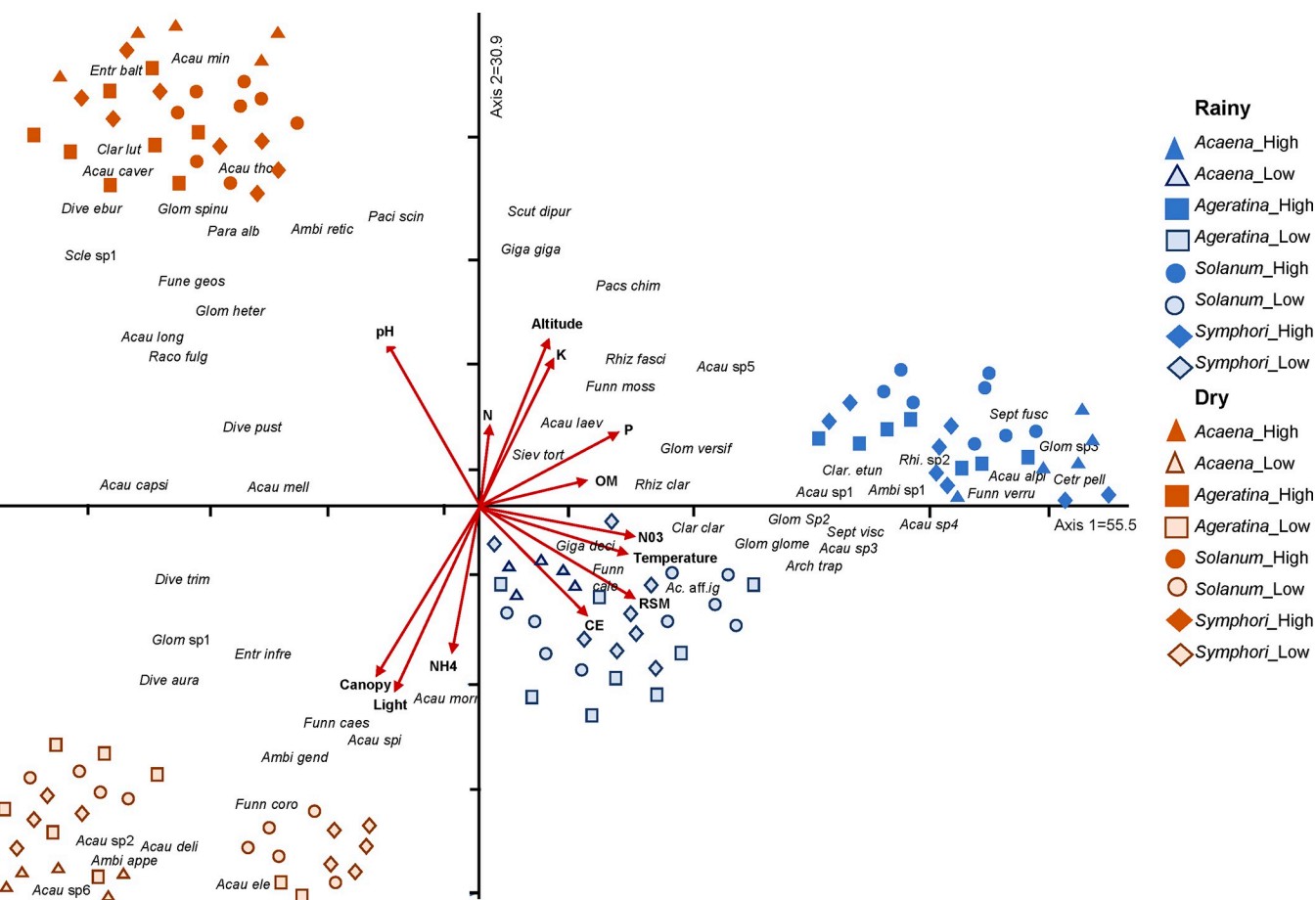

**Fig 6. Relationship between biotic and abiotic factors.** Canonical Correspondence Analysis (CCA) showing the relationship between arbuscular mycorrhizal fungi (AMF) community composition associated to rhizosphere soil of four host plant species and the abiotic factors (red arrows) in the *Abies religiosa* forest of the Magdalena river basin, Mexico City, Mexico. N = total nitrogen, pH = soil pH, MO = soil organic matter, CE = soil electrical conductivity, K = soil potassium. P = soil available phosphorus. $NH_4^+$ = ammonium, $NO_3^-$ = nitrate, RSM = relative soil moisture. Blue dots correspond to the rainy season, orange dots to the dry season. High = higher altitude and low = lower altitude. The different shapes are related to the plant species and the different colors are related to the season (blue = rainy and orange = dry). Where the species name is composed as follows: Plant genus_altitude (High = altitude and Low = altitude).

filters are the host plant identity, the interactions among the AMF species and, abiotic factors including edaphic and microclimatic.

Of the total AMF species richness recorded in our study, most of them were shared by all the host plant species (generalists). The recorded richness is significantly higher than previous reports for other temperate forests in Mexico [81]. The studied SV plant species showed differences in their AMF community composition, richness, and abundance, suggesting that host plant identity contributes to AMF community structuring. This may contribute to the tolerance of SV plant species to anthropogenic disturbances as the diversity of arbuscular mycorrhizal associations may provide increased resistance to environmental changes [82].

Assemblage processes of AMF communities in the soil and rhizosphere are not random [43]. We found that host plant species are associated with a set of locally present AMF species [44,45]. This shows that all host plant species (except *Acaena elongata*) were associated with a particular AMF species cluster, with some exclusive and most of them shared species, forming a specific and unique assemblage of AMF (S1 Table). Almost 25% of the AMF species were exclusive to at least one host plant species. This indicates that each host plant species

contributes to the maintenance of soil AMF species diversity that may be important during ecological succession contributing to the maintenance of a soil memory that modulates ecological interactions [83].

The host plant with the highest total AMF richness was *Symphoricarpos microphyllus*, followed by *Solanum pubigerum*, *Ageratina glabrata*, and finally *Acaena elongata* (Table 3). While these richness trends are indicative of biodiversity, they do not account for variation in AMF species composition among host plant individuals as shown in the NMDS analysis (Fig 5). This demostrate the importance of recording AMF composition and abundance. The identified patterns in AMF communities associated with host plant species, have previously been explained mainly by abiotic factors, but this study provides evidence of the importance of also considering the biological filters.

Relative to the biological filters, the four SV plant species are phylogenetically unrelated and have different biological traits such as reproductive phenology patterns, fruit types, and likely root architecture and exudate profiles, which have been proposed as drivers of AMF community composition in the rhizosphere [84,85]. For example, phenological stages are associated with different soil nutrient requirements, which modify the nutrient microenvironment of the rhizosphere and the plant-AMF relationship [84]. Interestingly, *Solanum pubigerum* and *Symphoricarpos microphyllus* show their highest fruit production and AMF diversity during the dry season, while for *Ageratina glabrata* this occurs during the rainy season [52]. Thus, AMF species diversity is influenced by the reproductive phenology of the host plants.

There is a variation in the host plant root architecture and the root exudate profiles of the studied species at the Family level. For example, Solanaceae are characterized by alkaloid glycosides [86]; Asteraceae by terpenoids, pyrrolizidine alkaloids and lactone sesquiterpenes [87]; Caprifoliaceae produce monoterpene glycosides [88,89], Rosaceae produce tannins, alkaloids and gallic acids [90]. Therefore, our results support the idea that all these biological traits in the four plant species contribute to the structuring of their AMF communities. Further studies explaining the relationship between a wide array of biological traits of plant species with their AMF species assemblage are needed.

Interactions between AMF species in the soil rhizosphere of host plants are a poorly understood biological filter. As shown by the cooccurrence analysis, most of the interactions between AMF species were negative, meaning that one species excluded others. This may indicate that the most competitive species have biological traits (e.g. mycelium length, spore production, fungal metabolites) that exclude other AMF species. On the other hand, the exclusion mechanisms of the remaining AMF species (those with a random association) may be absent and their ability to colonize the host roots depends on their broad tolerance or signal sensitivity to host exudates as well as on their broad tolerance to edaphic factors. The random AMF species were the most abundant in both seasons, altitudes and host plant species, i.e. generalist species, therefore these species have a wider host niche and contribute to the observed overlap values (Table 3). This is confirmed by the network interaction analysis and the CCA. While the AMF species with a positive relationship are exclusive to at least one host plant species. Our results suggest that the interactions between AMF species should be considered to understand their distribution and ecological niche, where some species have strong and specialized relationships, while others are more dependent on the competition between AMF species and edaphic factors.

Recent evidence suggests that AMF use intricate and complex sets of traits that maintain a unique genomic diversity among soil microorganisms, allowing them to adapt to different environmental conditions [91]. The interaction between different AMF species in the rhizosphere raises an important question: Is there active exclusion between them, or could colonization by a particular species inhibit the presence of others by altering secondary metabolites of

the host plant? This suggests that mycorrhizal associations may be influenced not only by mutualistic compatibility with their host plant, but also by competitive mechanisms that regulate or facilitate certain interactions between AMF species. Considering that host plant AMF communities contained 90% or more generalist AMF species and that 43% of all interactions between AMF species were negative (implying competitive exclusion), we suggest that the structuring of host AMF communities depends more on interactions between AMF species than on host traits that modulate AMF presence in the rhizosphere.

Simultaneously with the biological filters (host plant and AMF species interactions) AMF community structure is also modulated by the edaphic and environmental factors that vary among seasons and altitudes as shown in the NMDS and CCA analysis. The β-diversity is related to seasonal changes in abiotic factors. Our results are consistent with the hypothesis that the community structure of AMF is also influenced by seasonality and altitude.

It is evident that seasonal and altitudinal variations affect the abiotic factors that simultaneously modulate AMF species abundances in the rhizosphere of host-plant species, as shown b CCA analysis. *Acaena elongata*, *Ageratina glabrata* and *Solanum pubigerum* showed the highest AMF abundance during the dry season, these patterns could be a result of the changes in abiotic factors. The decrease in RSM, canopy openness and subsequent amount of light during the dry season stimulates the sporulation of some AMF species [92]. On the other hand, *Symphoricarpos microphyllus* showed a higher spore abundance during the rainy season, when there is an increase in RSM and temperature, a higher OM content, and higher P, N availability [93].

Species richness and spore abundance of AMF are higher at the low altitude than at the high altitude, where colder temperatures reduce sporulation [85]. In contrast, warmer temperatures and increased light at the low altitude stimulate sporulation [94]. Nutrient availability also varies with altitude and affects the AMF community, with P limitation at the low altitude driving mechanisms to enhance plant nutrient uptake and spore production [49]. These findings highlight the need to consider season and altitude when analyzing the factors influencing AMF community composition in temperate forests.

The functional traits of AMF species explain their interactions with plants and their adaptive strategies in different environments. *Acaulospora*, the most abundant genus (67%) in the rhizosphere of the studied plant species, thrives in soils with slightly acidic pH and high OM content, which are common in temperate forests [81]. This suggests that certain edaphic factors influence the ecological niche and community structure of AMF species [85,95]. The ability of *Acaulospora* to tolerate disturbance and persist through spore dormancy highlights its role in maintaining soil quality in forest ecosystems [96,97]. In contrast, Gigasporaceae and Glomeraceae were less abundant (<0.1% and 3.7%, respectively). Gigasporaceae, associated with a competitive life strategy, sporulate late in the dry season, explaining their low abundances [18]. Glomeraceae, known for their ruderal strategy, were more abundant in areas with frequent disturbance, and were less abundant in more conserved areas [18,19]. This highlights how functional diversity and niche differentiation structure AMF communities under different environmental conditions.

According to the variation partition analysis (VPA), the interactions between the biological and edaphic filters are key to understanding the AMF community structure, and therefore their classical separate evaluation does not provide an integral vision of the host-AMF association. This suggests that the interactions between filters are more important than the filters themselves, as confirmed in the CCA analysis.

## Conclusions

In conclusion, our study has shown that a complex interplay of ecological filters, in particular the biological identity of host plants, the interactions among AMF species, and the variation in

edaphic factors, significantly shape the composition and structure of AMF communities within the secondary vegetation of a temperate forest. We have shown how these filters, together with environmental variation due to seasonal changes and altitudinal floors, critically modulate AMF composition. This has improved the understanding of the taxonomic and functional diversity within AMF species assemblages and contributes to our broader ecological knowledge, particularly regarding the mechanisms that maintain plant-soil-microbe interactions in forest ecosystems. Furthermore, our study shows that there is an urgent need to emphasize the importance of recognizing the role of SV plant species in maintaining local AMF diversity.

## Supporting information

**S1 Fig. AMF species from secondary vegetation at *A. religiosa* forest of the Magdalena river basin.** a) *Funneliformis caledonius*. b) *Glomus glomerulatum*. c) *Glomus spinuliferum*. d) *Rhizophagus clarus*. e) *Rhizophagus fasciculatus*. f) *Acaulospora spinosa*. g) *Gigaspora decipiens*. h) *Racocetra fulgida*. i) *Scutellospora dipurpurescens*. j) *Sieverdingia tortuosa*. k) *Pacispora chimonobambusae*. l) *Ambispora appendicula*. m) *Ambispora gerdemannii*. n) *Ambispora reticulata*. o) *Entrophospora infrequens*, Bars = 50 μm.
(TIFF)

**S2 Fig. Rarefaction curves and extrapolation of AMF diversity in the rhizosphere of secondary vegetation in the *Abies religiosa* forest of the Magdalena river basin, Mexico.** *Acaena elongata* (orange), *Ageratina glabrata* (blue), *Solanum pubigerum* (pink) and *Syphoricarpos microphyllus* (purple). a) Comparison of the effective species diversity (0D), b) typical species (1D) and c) dominant species (2D). The solid line indicates interpolation, the dotted line indicates extrapolation, and the shaded area represents the 95% confidence interval for each plant species.
(TIFF)

**S3 Fig. AMF Beta diversity.** Non-metric multidimensional scaling showing the ordination of sampled individuals (dots) of each host plant species in the rainy and dry season and their location at high and low altitudes according to their AMF species composition, in the *Abies religiosa* forest of the Magdalena river basin, Mexico City, Mexico. The different shapes are related to the host plant species and altitude, and the different colors are related to the season (bluish green = rainy and orange = dry).
(TIFF)

**S4 Fig. Variance partitioning in arbuscular mycorrhizal fungi (AMF) communities, illustrating the contribution of biotic and abiotic factors and their interaction.**
(TIFF)

**S1 Table. Recording of arbuscular mycorrhizal fungi.** List of the arbuscular mycorrhizal fungi species registered in the rhizosphere soil of four plant species of secondary vegetation: *Acaena elongata*, *Ageratina glabrata*, *Solanum cervantesii* and *Symphoricapos microphyllus* during the rainy and the dry seasons in the *Abies religiosa* forest of the Magdalena river basin, Mexico City, Mexico.
(DOCX)

**S2 Table. Statistical summary of the Monte Carlo permutation test.** Analysis conducted for Canonical Correspondence Analysis (CCA) between edaphic-environmental factors and AMF community structure of four plant species in the *Abies religiosa* forest of the Magdalena river basin, Mexico City, Mexico.
(DOCX)

## Acknowledgments

We are grateful to Ariadna Peralta Valencia, and Irene Sánchez Gallén for their help in the field work and technical support in laboratory. To Marco Romero-Romero for his help in figure edition. This research is part of the doctoral studies of Y. Vázquez-Santos in the Posgrado en Ciencias Biológicas of the Universidad Nacional Autónoma de México. Y. Vázquez-Santos acknowledges Consejo Nacional de Humanidades, Ciencias y Tecnologías (CONHACyT)-Mexico (No. 818569) for scholarships to pursue her doctoral degree. We also thank Project IN211118.

## Author Contributions

**Conceptualization:** Yasmin Vázquez-Santos, Silvia Castillo-Argüero, Francisco Javier Espinosa-García, Noé Manuel Montaño.

**Data curation:** Yasmin Vázquez-Santos.

**Formal analysis:** Yasmin Vázquez-Santos.

**Investigation:** Silvia Castillo-Argüero.

**Methodology:** Yasmin Vázquez-Santos, Silvia Castillo-Argüero, Yuriana Martínez-Orea, Laura V. Hernández-Cuevas.

**Supervision:** Francisco Javier Espinosa-García, Noé Manuel Montaño, Yuriana Martínez-Orea.

**Validation:** Laura V. Hernández-Cuevas.

**Writing – original draft:** Yasmin Vázquez-Santos, Silvia Castillo-Argüero, Francisco Javier Espinosa-García, Noé Manuel Montaño, Yuriana Martínez-Orea.

**Writing – review & editing:** Yasmin Vázquez-Santos, Silvia Castillo-Argüero, Francisco Javier Espinosa-García, Noé Manuel Montaño, Yuriana Martínez-Orea, Laura V. Hernández-Cuevas.

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
