## [Decision Letter · Decision Letter 0]

3 Sep 2024

PONE-D-24-22732Ecological filters shape arbuscular mycorrhizal fungal communities in the rhizosphere of secondary vegetation species in a temperate forestPLOS ONE

Dear Dr. Castillo-Argüero,

Thank you for submitting your manuscript to PLOS ONE. After careful consideration, we feel that it has merit but does not fully meet PLOS ONE’s publication criteria as it currently stands. Therefore, we invite you to submit a revised version of the manuscript that addresses the points raised during the review. Please submit your revised manuscript by Oct 18 2024 11:59PM. If you will need more time than this to complete your revisions, please reply to this message or contact the journal office at plosone@plos.org. Please include the following items when submitting your revised manuscript:A rebuttal letter that responds to each point raised by the academic editor and reviewer(s). You should upload this letter as a separate file labeled 'Response to Reviewers'.A marked-up copy of your manuscript that highlights changes made to the original version. You should upload this as a separate file labeled 'Revised Manuscript with Track Changes'.An unmarked version of your revised paper without tracked changes. You should upload this as a separate file labeled 'Manuscript'.

We look forward to receiving your revised manuscript.

Kind regards,

Erika Kothe

Academic Editor

PLOS ONE

Journal Requirements:

5. We note that Figure 1 in your submission contain map/satellite images which may be copyrighted. All PLOS content is published under the Creative Commons Attribution License (CC BY 4.0), which means that the manuscript, images, and Supporting Information files will be freely available online, and any third party is permitted to access, download, copy, distribute, and use these materials in any way, even commercially, with proper attribution. For these reasons, we cannot publish previously copyrighted maps or satellite images created using proprietary data, such as Google software (Google Maps, Street View, and Earth). For more information, see our copyright guidelines: http://journals.plos.org/plosone/s/licenses-and-copyright.

Additional Editor Comments:

The work is robust and adds to current knowledge. However, the main findings are diluted by the overly lengthy results and discussion. A major re-draftin is necessary following the advise of the expert reviewer.

Reviewers' comments:

Reviewer's Responses to Questions

**Comments to the Author**

1. Is the manuscript technically sound, and do the data support the conclusions?

Reviewer #1: Yes

2. Has the statistical analysis been performed appropriately and rigorously? 

Reviewer #1: Yes

3. Have the authors made all data underlying the findings in their manuscript fully available?

Reviewer #1: Yes

4. Is the manuscript presented in an intelligible fashion and written in standard English?

Reviewer #1: Yes

5. Review Comments to the Author

Reviewer #1: I have revised the manuscript PONE-D-24-22732. It is very interesting by considering the AMF assemblage in plant species from secondary forest. It presents a robust analysis regarding AMF from temperate forest. Regarding this manuscript, I had specific questions that are written below:

What is the main question addressed by the research?

The authors analyzed the relationship between biotic and abiotic factors and the AMF community assembly in the rhizosphere of four secondary vegetation (SV) plant species in a temperate forest.

What parts do you consider original or relevant for the field?

The results and discussion are original and relevant to the fields of soil microbiology, and soil science. However, the authors could use more descriptive subsections by reducing the lenght of the results and discussion for being more punchy.

What specific gap in the field does the paper address?

The results of this manuscript effectively enhance the understanding of AMF by considering season, altitude, host-plants, and their interaction on AMF community composition.

What does it add to the subject area compared with other published material?

It adds an interesting dataset following a robust statistical analysis (GLM, Permanova, and CCA).

What specific improvements should the authors consider regarding the methodology? What further controls should be considered?

The authors could reduce the lenght of the results and discussion for being more punchy. Both section are too long and hard to follow in the current version.

Please describe how the conclusions are or are not consistent with the evidence and arguments presented. Please also indicate if all main questions posed were addressed and by which specific experiments.

The conclusions are consistent with the aim of the study. All questions were addressed in the results and discussion sections.

Are the references appropriate?

Yes, they are appropriate. However, the author could reduce the number of cited works.

Please include any additional comments on the tables and figures and the quality of the data.

The tables and figures follow the authors' guidelines. However, all figures have low resolution and need to be improved. They are interesting, but hard to read and understand in the current version. The tables provide enough information and are well-presented.

6. PLOS authors have the option to publish the peer review history of their article (what does this mean?). If published, this will include your full peer review and any attached files.

Reviewer #1: **Yes: **Tancredo Souza

---

## [Author Response · Author response to Decision Letter 0]

9 Oct 2024

Mexico City, Mexico, september 23th, 2024

Emily Chenette

Editor-in-Chief

PLOS ONE

Please find attached the new and corrected version of our manuscript entitled "Ecological filters shape arbuscular mycorrhizal fungal communities in the rhizosphere of secondary vegetation species in a temperate forest", authored by Yasmin Vázquez-Santos, Silvia Castillo-Argüero, Francisco Javier Espinosa-García, Noé Manuel-Montaño, Yuriana Martínez-Orea and Laura V. Hernández-Cuevas, which was accepted with changes in your Journal.

We attended all the suggestions and corrections to improve the manuscript quality.

Below we include the responses to the suggested changes (with some comments made by the reviewers and editor).

Journal Requirements:

Response: Following the PLOS ONE style templates, we made the style changes required by the journal.

Response: We have included this information in the New Methods section.

- The values used to build graphs

Response: We have included in the manuscript and in the supplementary information all the information necessary to replicate our results.

Response: We agree with all disclosure statements regarding data availability.

5. We note that Figure 1 in your submission contain map/satellite images which may be copyrighted. All PLOS content is published under the Creative Commons Attribution License (CC BY 4.0), which means that the manuscript, images, and Supporting Information files will be freely available online, and any third party is permitted to access, download, copy, distribute, and use these materials in any way, even commercially, with proper attribution. For these reasons, we cannot publish previously copyrighted maps or satellite images created using proprietary data, such as Google software (Google Maps, Street View, and Earth). For more information, see our copyright guidelines:http://journals.plos.org/plosone/s/licenses-and-copyright.

Response: We had asked for the permission to publish this figure to the author and included it as an attached material.

Additional Editor Comments

Response: We have verified that the manuscript meets all publication requirements.

7. What parts do you consider original or relevant for the field?

The results and discussion are original and relevant to the fields of soil microbiology, and soil science. However, the authors could use more descriptive subsections by reducing the lenght of the results and discussion for being more punchy. 

What specific improvements should the authors consider regarding the methodology? What further controls should be considered?

The authors could reduce the lenght of the results and discussion for being more punchy. Both sections are too long and hard to follow in the current version.

Response: We have made great efforts to reduce the length of the results and discussion sections, and we have also emphasized our relevant findings on AMF species assemblages and the filters involved. We believe that our new version is easier to follow.

8. Are the references appropriate? Yes, they are appropriate. However, the author could reduce the number of cited works.

Response: We reduced the number of cited works.

9. Please include any additional comments on the tables and figures and the quality of the data.

The tables and figures follow the authors' guidelines. However, all figures have low resolution and need to be improved. They are interesting, but hard to read and understand in the current version. The tables provide enough information and are well-presented.

Response: We improved the resolution of all figures.

---

## [Decision Letter · Decision Letter 1]

4 Nov 2024

Ecological filters shape arbuscular mycorrhizal fungal communities in the rhizosphere of secondary vegetation species in a temperate forest

PONE-D-24-22732R1

Dear Dr. Castillo-Argüero,

We’re pleased to inform you that your manuscript has been judged scientifically suitable for publication and will be formally accepted for publication once it meets all outstanding technical requirements.

Kind regards,

Erika Kothe

Academic Editor

PLOS ONE

Additional Editor Comments (optional):

Reviewers' comments:

Reviewer's Responses to Questions

**Comments to the Author**

1. If the authors have adequately addressed your comments raised in a previous round of review and you feel that this manuscript is now acceptable for publication, you may indicate that here to bypass the “Comments to the Author” section, enter your conflict of interest statement in the “Confidential to Editor” section, and submit your "Accept" recommendation.

Reviewer #1: All comments have been addressed

2. Is the manuscript technically sound, and do the data support the conclusions?

Reviewer #1: Yes

3. Has the statistical analysis been performed appropriately and rigorously? 

Reviewer #1: Yes

4. Have the authors made all data underlying the findings in their manuscript fully available?

Reviewer #1: Yes

5. Is the manuscript presented in an intelligible fashion and written in standard English?

Reviewer #1: Yes

6. Review Comments to the Author

Reviewer #1: The authors have improved all sections in the current version of the ms. I have no further comments. Congratulations.

7. PLOS authors have the option to publish the peer review history of their article (what does this mean?). If published, this will include your full peer review and any attached files.

Reviewer #1: **Yes: **Tancredo Souza

---

## [Editor Report · Acceptance letter]

14 Nov 2024

PONE-D-24-22732R1 

PLOS ONE

Dear Dr. Castillo-Argüero, 

I'm pleased to inform you that your manuscript has been deemed suitable for publication in PLOS ONE. Congratulations! Your manuscript is now being handed over to our production team.

Kind regards, 

on behalf of

Prof. Dr. Erika Kothe 

Academic Editor

PLOS ONE